# Some Clues about Enzymes from Psychrophilic Microorganisms

**DOI:** 10.3390/microorganisms10061161

**Published:** 2022-06-06

**Authors:** Roberta Rapuano, Giuseppe Graziano

**Affiliations:** Dipartimento di Scienze e Tecnologie, Università del Sannio, Via Francesco de Sanctis snc, 82100 Benevento, Italy; rrapuano@unisannio.it

**Keywords:** psychrophilic microorganisms, psychrophilic enzymes, conformational stability, enzymatic activity, structural flexibility, vibrational normal modes

## Abstract

Enzymes purified from psychrophilic microorganisms prove to be efficient catalysts at low temperatures and possess a great potential for biotechnological applications. The low-temperature catalytic activity has to come from specific structural fluctuations involving the active site region, however, the relationship between protein conformational stability and enzymatic activity is subtle. We provide a survey of the thermodynamic stability of globular proteins and their rationalization grounded in a theoretical approach devised by one of us. Furthermore, we provide a link between marginal conformational stability and protein flexibility grounded in the harmonic approximation of the vibrational degrees of freedom, emphasizing the occurrence of long-wavelength and excited vibrations in all globular proteins. Finally, we offer a close view of three enzymes: chloride-dependent α-amylase, citrate synthase, and β-galactosidase.

## 1. Introduction

Despite our conception, Earth is a rather cold place. Temperatures below 10 °C characterize about 80% of Earth’s biosphere [1], of which 70% is made up of oceans (see Figure 1). The remaining percentage includes the polar regions, the mountainous regions, the mesosphere, and the stratosphere. A small contribution is also given by man and therefore by artificial niches attributable to the use of refrigerators and freezers [2,3]. Microorganisms have been discovered capable of living in conditions of extremely low temperatures compared to what we define “ambient temperature” [4]. Organisms living in these environments are known as psychrophiles; some of them reach maximum growth even at temperatures below 0 °C. Psychrophiles are mostly localized in marine ecosystems, particularly in the abysses, where the water temperature is permanently cold. Both cell lines and bacterial cultures can be stored in liquid nitrogen, where the temperature is −196 °C. Despite this, the temperature recordings carried out in monitoring the activities of these psychrophilic microorganisms did not report values below −20 °C [5]. There is an increasing interest in studying psychrophilic microorganisms to discover their adaptation mechanisms. The extreme environments are increasingly seen as a source of new microbes and biomolecules that could have potential biotechnological and biomedical applications [6,7,8]. For example: (a) enzymes active in the cold have been identified in microbial cultures from marine environments of polar regions [9,10]; (b) antibiotic-producing bacteria have been isolated from permafrost soils in the upper Arctic regions [11]. It is evident that psychrophiles have developed evolutionary defense mechanisms that support their survival in such inhospitable environments. The adaptation mechanisms allow these organisms to effectively cope with cold. Furthermore, physiological and genomic adaptations, such as the biosynthesis of cryoprotectants, changes in membrane composition and the functioning of enzymes, provide mechanisms to compensate for the kinetic and thermodynamic effects of low temperatures [12,13]. To maintain optimal membrane fluidity, for example, the cell must regulate membrane composition as well as the quantity and type of lipids (saturated or unsaturated). Since the membrane fluidity decreases as the temperature decreases, microorganisms show an increase in the ratio of unsaturated to saturated fatty acids. Furthermore, both the structure and functions of proteins are known to be influenced by temperature [14]. Thermophilic proteins are characterized by an increase in the content of ion-pairs, by the formation of higher-order oligomers and by a decrease in structural flexibility [15]. The latter seems to be the consequence of a decrease in surface loop length, optimization of electrostatic and van der Waals interactions, and amino acid substitutions aimed at increasing internal hydrophobicity [16,17,18].

On the other hand, the ability to withstand sub-zero temperatures relies on two strategies. The first one is to avoid the formation of ice in order to protect cells from freezing, synthetizing specific proteins called ice binding proteins, IBP [19,20]; the second one is to develop protective mechanisms to avoid damage during the thawing phase in case of still formed ice [5,21]. The proteins used in both processes are called “antifreeze” molecules and allow the freezing point of water to be lowered by more than 10 °C. The adaptation mechanisms of psychrophilic enzymes are different from those preferred by thermophilic ones [22]. The random thermal energy conferring structural flexibility and mobility to a protein chain is small at low temperature, nevertheless, the enzyme activity has to be as high as necessary to sustain the microorganism’s life. Thus, it is expected that psychrophilic enzymes have a greater structural flexibility to increase the catalytic rate constant, and a higher degree of conformational complementarity with substrates to allow a decrease in activation energy, when compared to mesophilic and thermophilic homologues [23]. In order to shed light on these fundamental issues and to be sure that significant differences emerge, several structural comparative studies have been performed on enzyme families belonging to psychrophiles, mesophiles and thermophiles [24,25,26,27,28,29,30,31,32,33,34,35]. Even though adaptation strategies are family-specific, the following common rules hold for psychrophilic proteins: (a) the structural differences between the folded states are small and subtle, notwithstanding large differences in thermal stability; (b) there are several charged residues on the surface of psychrophilic enzymes to increase solubility and flexibility, not the conformational stability; (c) the interior packing, on the average, does not change; (d) there is a higher percentage of tiny residues (i.e., Ala, Gly, Ser, and Thr), whose small side chains make psychrophilic enzymes less compact and more flexible than the mesophilic and thermophilic counterparts. In the present study we provide: (a) a view of metabolism adaptation to cope with cold environments; (b) a study of the thermodynamic characteristics of globular proteins and their conformational stability in general, with a specific application of the so-called geometric model to psychrophilic proteins; (c) an analysis of enzymatic activity with a specific focus on psychrophilic enzymes and closer inspection of the adaptation mechanisms of α-amylase, citrate synthase and β-galactosidase. It is worth noting that enzymes from psychrophilic microorganisms are largely used in industrial applications due to their high catalytic activity at low temperature [36,37]. For example, psychrophilic lipases play an important role in allowing a reduced washing temperature in laundry industry and so a significant energy saving; psychrophilic xylanases are a key ingredient to improve the bread quality for their ability to degrade the beta-1,4-xylan; psychrophilic α-amylases are largely used in food industry, for example to avoid haze formation in juice, and in baking industry processes [38,39,40]. Moreover, a DNA polymerase I, isolated from the psychrophilic *Psychromonas grahamii*, has very good catalytic activity at low temperature and is of potential interest for sequencing companies [41]. Readers interested in industrial and biotechnological applications of psychrophilic enzymes are referred to reviews specifically devoted to this topic [33,42,43,44,45].

## 2. Metabolism Adaptation

A Biokinetic Spectrum for Temperature, recently determined [46], represents the distribution of microorganism growth rate as a function of the environmental temperature, and covers a temperature range greater than 120 °C, emphasizing that all the biosphere hosts living organisms. This Biokinetic Spectrum for Temperature shows the maximum growth rate, at each temperature, for any life form on Earth, and it vividly clarifies that the growth rate is very small at low temperature [47,48]. Therefore, psychrophiles, the organisms capable of adapting to the coldest ecosystems on Earth and able to survive in conditions of low temperature, deserve great attention. The evolution of these organisms has made it possible to develop mechanisms that allow an adjustment of metabolic activities in adverse situations as, for example, the slowdown of enzymatic activities or modification in the function of proteins involved in metabolic cycles [7,49]. Cold shock largely influences cellular processes including: protein synthesis, absorption of nutrients, cell growth and membrane fluidity [50,51]. In very cold environments it is necessary to develop adaptive characteristics at the level of functionality and cellular structure [52]. Among these, one of the most important aspects concerns the alterations in expressed genes and, consequently, in proteins involved in specific metabolic pathways [26,53,54]. Cold shock leads to an up-regulation of genes coding for cold shock proteins (CSP). These proteins are expressed in mesophilic organisms when exposed to cold, while, in psychrophilic microorganisms, they are constitutively expressed and are classified as cold acclimatization proteins (CAP) [7]. Several studies have revealed that some psychrophilic communities are located in small ice pockets with high saline concentrations, rich in organic material that supports the population growth [55]. In some cases, the microorganism metabolism has to be compatible with temperatures below the freezing point of water [25]. Recent studies have shown that the induction of thermal stress drastically reduces the metabolic activity and that only a small portion of proteins is able to adapt to low temperatures [56]. Usually, in very cold environments, reactive oxygen species (ROS) accumulation is more pronounced. It has been found that genes coding for antioxidant enzymes, such as catalase and superoxide dismutase (SOD), are more expressed in psychrophiles [39,57]. Linked to an evident slowing down of the biosynthetic pathways, this is one of the regulatory mechanisms through which these organisms are able to avoid damage due to ROS accumulation [3,58]. In psychrophiles, the genes coding for proteins involved in flagellar motility and energy metabolism are down-regulated [59,60], while it has been evidenced an up-regulation in pathways involved in arginine degradation, ROS neutralization, and biosynthesis of proline, polyamines, and unsaturated fatty acids [61]; this is highlighted in Figure 2. Furthermore, a marked reduction in the expression levels of oxidative stress-related proteins was recorded in *Pseudoalteromonas haloplanktis* under cold stress growth conditions [9]. In addition to the repression of ROS-generating oxidative metabolic pathways, it was reported that the production level of several osmo-protectants [62], such as trehalose, increased [61]. The production of trehalose represents an adaptation mechanism in protecting microorganisms against a wide variety of lethal conditions [63,64]. It has been shown that, in cold stress, some cytoprotective polyamines are produced to stabilize nucleic acids and neutralize ROS [65]. Some of these polyamines come from arginine degradation [66], accomplished by the arginine decarboxylase enzyme in psychrophilic *Pseudomonas helmanticensis* [61].

Following thermal shock, organisms trigger adaptive cryo-protection changes to reduce cell permeability and mitochondrial swelling [67]. The latter mitochondrial swelling occurs in mesophiles, but is not observed in psychrophiles. Another cryo-protection mechanism is the accumulation of proline. Proline is a membrane stabilizing agent, making direct interactions with phospholipids and rendering them less sensitive to low temperatures [68].

## 3. Thermodynamic Features

To gain the right perspective on enzymes extracted from psychrophilic microorganisms, it is necessary and useful to account for the main results obtained about the conformational stability of globular proteins in general. Circular dichroism (CD) and differential scanning calorimetry (DSC) measurements of globular protein thermal stability allowed the determination of the denaturation temperature, T_d_, the denaturation enthalpy change, ΔH_d_(T_d_), and the denaturation heat capacity change, ΔC_p,d_, for a large number of globular proteins and their mutant forms [69,70]. Normally, the temperature-induced denaturation of small globular proteins (i.e., single domain globular proteins; a structural domain should consist of less than 200 residues [71]) is a reversible and cooperative process, corresponding to a first-order phase transition between the native and denatured macro-states (N-state and D-state, respectively); it is an all-or-none, cooperative process [72]. The knowledge of the three thermodynamic parameters listed above, assuming ΔC_p,d_ temperature-independent, makes it possible to calculate the denaturation Gibbs free energy change, ΔG_d_ = G(D-state) − G(N-state), as a function of temperature, using the Gibbs-Helmholtz equation. The trend of ΔG_d_ as a function of temperature is called the protein stability curve [73]. In the temperature range where ΔG_d_ is positive, the N-state is stable; in the temperature range where ΔG_d_ is negative, the D-state is stable; the temperature at which ΔG_d_ = 0 corresponds to the denaturation temperature of the protein, where the two states have the same thermodynamic stability (i.e., the equilibrium constant K_d_ = [D-state]/[N-state] = exp[−ΔG_d_(T_d_)/RT_d_] = 1, and so the two macro-states are equally populated). In general, since ΔC_p,d_ is a large positive quantity, the protein stability curve has a parabolic shape with two temperatures at which ΔG_d_ = 0, corresponding to the unusual cold denaturation and the common hot denaturation, respectively. The former is unusual because: (a) it occurs on lowering the temperature from ambient conditions, and is characterized by a negative enthalpy change and a negative entropy change; (b) in the case of stable globular proteins, it occurs well below 0 °C, and so is not easy to study. Nevertheless, cold denaturation has directly been detected using DSC, CD and NMR measurements on destabilized mutant forms, or on wild-type proteins in destabilizing conditions, such as aqueous urea solutions [74,75]. The common hot denaturation occurs on increasing the temperature, is characterized by large positive enthalpy and entropy changes, and is driven by the large gain in conformational entropy of the polypeptide chain associated with unfolding. The T_d_ values of mesophilic proteins are spread over a wide temperature range with an average at 60 °C, well above the living temperature of mesophiles [76]. Similarly, for the small number of psychrophilic proteins whose temperature-induced denaturation has carefully been investigated, the T_d_ values are clustered around 40°C [26], well above the living temperature of psychrophiles. It seems that the N-state of both psychrophilic and mesophilic globular proteins is more stable than it would be strictly necessary. A survey over experimental data of a large set of globular proteins [77] has emphasized, in line with previous analyses [76,78], that the maximum of the stability curve is located at T_max_ = 284 ± 19 K, and ΔG_d_(T_max_) ≈ 20–40 kJ mol^−1^ for a protein of about 100 residues. These two heuristic findings indicate that: (a) the maximal thermodynamic stability occurs around room temperature, regardless of the T_d_ value, in all probability because the amino acid content of all proteins is similar, on the average; (b) the N-state is only marginally more stable than the D-state because a delicate balance between large stabilizing and large destabilizing interactions holds, and depends on temperature [69,76,78]. It is possible to calculate the stability curve for a mesophilic model protein and a psychrophilic model protein, both consisting of 100 residues, exploiting the relationships for mesophilic proteins reported by Robertson & Murphy [76]. For the psychrophilic model protein, the T_d_ and ΔH_d_(T_d_) numbers have suitably been lowered with respect to those of Robertson & Murphy, whereas ΔC_p,d_ has not been modified in view of the close structural similarity between a mesophilic enzyme and its psychrophilic homologue [27]. Specifically: (i) T_d_ = 333 K, ΔH_d_(T_d_) = 292 kJ mol^−1^ and ΔC_p,d_ = 5.8 kJ K^−1^ mol^−1^ for the mesophilic model; (ii) T_d_ = 313 K, ΔH_d_(T_d_) = 176 kJ mol^−1^ and ΔC_p,d_ = 5.8 kJ K^−1^ mol^−1^ for the psychrophilic model. The two calculated stability curves are shown in Figure 3; they practically possess the same T_max_ ≈ 285 K, and ΔG_d_(T_max_) is around 8 kJ mol^−1^ for the psychrophilic model, and around 21 kJ mol^−1^ for the mesophilic model (note that in both cases cold denaturation occurs at very low temperatures). The two stability curves shown in Figure 3 closely resemble those obtained by Feller and colleagues for a psychrophilic α-amylase and its mesophilic counterpart [79]. It is clear that the knowledge of the macroscopic thermodynamic quantities associated with denaturation does not provide molecular-level information about the interactions stabilizing the N-state and those stabilizing the D-state. Since the magnitude of ΔC_p,d_, ΔH_d,_ and ΔS_d_ linearly depends on the number of protein residues [76,77], a mean-field theoretical approach, neglecting the subtleties of amino acid sequence and the differences in secondary structure content, can be successful in clarifying some important features of the conformational stability of globular proteins. A geometric model is able to explain the existence of cold denaturation [80,81], the extra-thermal stability of thermophilic proteins on entropic grounds [82], and the high thermal stability of very small globular proteins [83]. We would like to apply such an approach to the conformational stability of psychrophilic proteins. The starting point of the approach is the recognition that the solvent-excluded volume effect plays a pivotal role in processes occurring in the liquid phase, that is a condensed state of matter. The presence of a solute molecule reduces the configurational space accessible to solvent molecules because no other molecule can occupy the space occupied by the solute molecule. Cavity creation at a fixed position in the liquid is the theoretical means to account for this basic fact [84,85]. Cavity creation at a fixed position, keeping constant temperature and pressure, leads to an increase in liquid volume by the partial molar volume of the cavity; this, however, does not remove the solvent-excluded volume effect. The solvent molecule centers cannot penetrate the solvent-accessible-surface of the cavity if the latter has to exist [80]. The solvent-excluded volume is given by the shell between the van der Waals surface of the cavity and its solvent-accessible-surface, and its measure can be provided by the water-accessible-surface-area, WASA, of the cavity [86]. This means that the size of the statistical ensemble of liquid configurations is markedly reduced by cavity creation: relevant configurations are solely the ones in which it does exist a cavity suitable to host the solute molecule. The latter are a tiny fraction of the total in the case of molecular-sized cavities [87]. A loss in translational entropy of water molecules in associated with a decrease in accessible configurational space [84]. It is interesting that the solvent-excluded volume effect can be quantified by a geometric measure such as the WASA of the cavity hosting the solute. In this respect, proteins are special, because, being polymers, can populate conformations characterized by largely different WASA values, causing different solvent-excluded volume effects.

It is the shape of the polypeptide chain to distinguish the conformations belonging to the N-state, from those belonging to the D-state. The simplest geometric choice is: a sphere can model the N-state, and a prolate spherocylinder can model the D-state. The two objects have equal V_vdW_, (i.e., the volume practically does not change on unfolding at P = 1 atm [88,89]), but different WASA (see Figure 4). The spherocylinder has larger WASA because it is well-known that, with a fixed volume, the sphere is the 3D object possessing the smallest surface area. As a consequence, the solvent-excluded volume experienced by water molecules will be different. The minimization of the loss in translational entropy is the reason why water molecules stabilize the N-state. Indeed, with a fixed cavity V_vdW_, the ΔG_c_ magnitude rises with cavity WASA [90]. Thus, it is possible to calculate the difference between ΔG_c_ to create in water a cavity suitable to host the prolate spherocylinder corresponding to the D-state, and ΔG_c_ to create in water a cavity suitable to host the sphere corresponding to the N-state. The ΔΔG_c_ = ΔG_c_(D-state) − ΔG_c_(N-state) contribution is always positive, stabilizing the N-state; it practically accounts for what is usually called the hydrophobic effect [91]. The large gain in conformational entropy upon denaturation destabilizes the N-state. It can be calculated by T⋅ΔS_conf_ = T⋅N_res_⋅ΔS_conf_(res), assuming equal and additive contributions by the chain residues. This rough approximation works well because it has been found that the denaturation entropy change ΔS_d_ is a linear function of N_res_ [76,77]; note that ΔS_conf_ is a large part of ΔS_d_.

Apart the two previous entropic contributions, it is necessary to account for an energetic term that differentiates the two macro-states; ΔE_a_ = [E_a_(D-state) − E_a_(N-state) + ΔE(intra)] is the difference in energetic interactions among the D-state and the N-state and surrounding water molecules, and the difference in intra-chain energetic interactions between the D-state and the N-state [81]. This energetic contribution should not be large because: (a) the H-bonds that peptide groups make with water molecules in the D-state are largely reformed as intra-protein H-bonds in the secondary structure elements of the N-state; (b) side chains able to make H-bonds usually form them both with water molecules in the D-state, and intra-molecularly in the N-state; a H-bond satisfaction principle holds [92]; (c) van der Waals attractions between protein groups and water molecules are regained as intra-protein contacts in the close-packed interior of the N-state. In particular, a complete balance between protein-water energetic attractions in the D-state and N-state, respectively, and the intra-protein energetic attractions implies that ΔE_a_ = 0. In this case, the ΔE_a_ quantity should not contribute to the overall Gibbs free energy balance; this situation does appear realistic. Considering experimental data for 115 different globular proteins [77], the denaturation enthalpy change ΔH_d_ proves to be zero at a temperature T_H_ = (277.5 ± 25) K, corresponding to the characteristic maximum density temperature of H_2_O [93]; for detailed explanations the reader is referred to [82,83]. On this basis, it is suitable to assume that: (a) the ΔE_a_ energetic term is temperature-independent; (b) ΔE_a_ = 0 for a mesophilic model protein; (c) ΔE_a_ < 0 for a psychrophilic model protein (i.e., energetic factors favor the D-state). This last point is reliable because structural comparative analyses have shown that the N-state interior of psychrophilic proteins possesses, on average, a smaller number of stabilizing energetic attractions (i.e., H-bonds and van der Waals contacts) with respect to mesophilic and thermophilic homologues [27,29]. Indeed, both Kanaya and co-workers on ribonuclease HI from the psychrotrophic bacterium *Shewanella oneidensis* MR-1 [94], and Feller and co-workers on the α-amylase from the Antarctic bacterium *P. haloplanktis* [95], have shown that, inserting in the cold-active protein some selected residues present in the mesophilic counterpart, thus allowing the formation of additional attractive interactions among side chains, the protein thermal stability increases significantly, and in an almost additive way. The latter feature is because each mutation affords a small contribution to the overall Gibbs free energy balance. The energetic destabilization mechanism should be the molecular basis of the higher structural flexibility attributed to psychrophilic enzymes. The approach provides the following formula for the denaturation Gibbs free energy change (note that the N-state is stable when ΔG_d_ is positive):ΔG_d_ = ΔΔG_c_ + ΔE_a_ − T⋅ΔS_conf_

It is important to underscore that the ΔE_a_ quantity does not correspond to ΔH_d_ (i.e., the quantity carefully measured in a DSC experiment), because the latter takes into account also the enthalpy contribution coming from the reorganization of water-water H-bonds associated with protein denaturation. This reorganization process, according to different theoretical arguments and routes [96,97,98,99,100], produces compensating enthalpy and entropy changes. Since the structures of psychrophilic proteins are closely similar to those of mesophilic and thermophilic homologues [27], the same sphere can model their N-state. On similar grounds, the same prolate spherocylinder can model their D-state. The geometric sizes are: the sphere has radius *a* = 15 Å, V_vdW_ = 14137 Å^3^ and WASA = 3380 Å^2^ (calculated using for the rolling sphere a radius of 1.4 Å [86]); the prolate spherocylinder has radius *a* = 6 Å, cylindrical length *l* = 117 Å, V_vdW_ = 14137 Å^3^ and WASA = 6128 Å^2^. These values should represent a protein of 138 residues [101]. The analytical formulas of classic scaled particle theory [90,102] allow the calculation of the ΔΔG_c_ function, using the experimental density of water [93]. The effective, temperature-independent, hard sphere diameter of water molecules has been fixed to σ (H_2_O) = 2.80 Å [81,84,103]. The conformational entropy contribution is calculated by fixing N_res_ = 138 and ΔS_conf_(res) = 19.0 J K^−1^ molres^−1^. The latter value is right for a mesophilic model protein [104], and should be suitable also for a psychrophilic model protein. For the latter, the energetic term has deliberately been fixed at ΔE_a_ = −13 kJ mol^−1^ to account for the energetic destabilization emerged from structural comparative analyses [27,28]. It is important to underscore that the magnitude of the ΔE_a_ energetic term is very small in comparison to that of the two entropic contributions; for instance, at 10 °C, ΔΔG_c_ ≈ 768 kJ mol^−1^ and T⋅ΔS_conf_ ≈ 743 kJ mol^−1^. This is because the energetic balance between the D-state and the N-state is always in action, even though it is not always complete. The calculated ΔΔG_c_ function (the black curve in Figure 5) has a parabolic temperature dependence and its magnitude does not depend on the mesophilic or psychrophilic origin of the protein since their N-states and D-states share the same geometric sizes.

The WASA increase associated with denaturation largely stabilizes the N-state. The ΔΔG_c_ increase with temperature is because the water density is almost temperature-independent, and the cavity surface has to contrast the kinetic energy of bombarding water molecules [105,106]. The T⋅ΔS_conf_ straight line (the red line in Figure 5) intersects the ΔΔG_c_ parabola in two points that represent the temperature of cold denaturation and the temperature of hot denaturation, respectively. The subtraction of the T⋅ΔS_conf_ straight line from the ΔΔG_c_ function leads to a stability curve (the black line in Figure 6) reliable for a mesophilic model protein, with T_d_(hot) ≈ 57 °C.

On the other hand, the same T⋅ΔS_conf_ straight line intersects the ΔΔG_c_ + ΔE_a_ function (the blue curve in Figure 5) at two points that are shifted at higher temperature for cold denaturation and lower temperature for hot denaturation. The obtainaed stability curve (the red line in Figure 6) is reliable for a psychrophilic model protein, with T_d_(hot) ≈ 42 °C and a lower ΔG_d_ at the maximum stability temperature. In particular, T_max_ ≈ 10 °C and ΔG_d_(T_max_) ≈ 25 kJ mol^−1^ for the mesophilic model protein, while T_max_ ≈ 10 °C and ΔG_d_(T_max_) ≈ 12 kJ mol^−1^ for the psychrophilic model protein. These numbers agree with experimental ones. It can be concluded that: (a) all globular proteins are characterized by a subtle balance between stabilizing and destabilizing terms [69,78]; (b) the inclusion of a destabilizing energetic term in the framework of the geometric theoretical approach leads to a model protein with psychrophilic features.

## 4. Enzymatic Adaptation

First Somero [107,108], and soon after Jaenicke [109] proposed that enzymes from microorganisms adapted to different environmental temperatures work in “corresponding states” conditions. Since the substrate-binding sites are strictly conserved from the structural point of view [26], and since to catalyze a given biochemical reaction the requested vibrations (i.e., structural movements) are always the same, a closely similar flexibility has to be associated with the psychrophilic enzyme at 5–20 °C, the mesophilic enzyme at 35–50 °C, and the thermophilic enzyme at 75–90 °C. The “corresponding states” hypothesis refers to the functioning of enzymes, not to their conformational stability. This hypothesis is reliable and has received some experimental confirmation [110]. It would be important to link the emerged marginal thermodynamic stability of the N-state with the significant structural fluctuations characterizing globular proteins. Such structural fluctuations can be described as vibrational normal modes: a molecule consisting of N atoms possesses 3N degrees of freedom, (3N − 6) of which are associated with vibrations, that can be treated as normal modes in the harmonic approximation. Clearly, the number of vibrational normal modes is very large also for a small globular protein. Their frequency distribution (i.e., the vibrational density of states, VDOS) should depend upon the molecular structure, and so should be different between the N-state and the D-state; more importantly, it is expected to be different among psychrophilic, mesophilic, and thermophilic enzymes. Actually, Song and colleagues [111] calculated the VDOS of 135 different globular proteins, starting from their crystallographic structures, and showed that it is universal, on average. The calculated VDOS is characterized by a large number of normal modes with frequency below 300 cm^−1^. Since RT = 2.5 kJ mol^−1^ at 300 K, and the latter energy corresponds to a frequency of 208 cm^−1^, a large number of vibrational normal modes are populated and excited at room temperature. This means that the N-state of globular proteins is highly dynamic and fluctuating, notwithstanding its ability to preserve a single average structure. Molecular vibrations, especially those involving the substrate-binding site, are necessary to perform catalytic activity [112]. Those useful for the catalytic function possess high frequency (i.e., greater than 1000 cm^−1^), and small wavelength, and should be localized in the substrate-binding region [113]. Note that a mechanism of energy transfer from the accessible normal modes (those with frequency smaller than 300 cm^−1^) to the un-accessible ones (those with a frequency greater than 1000 cm^−1^) has to be operative and should merit special attention. These functional vibrations should have the same degree of excitation at the “corresponding” working temperature of the psychrophilic, mesophilic and thermophilic enzyme (i.e., should possess a similar excess of energy to speed up the reaction). It should be clear that, if the enzymatic activity is used to interrogate and measure protein flexibility at room temperature, psychrophilic enzymes do appear flexible, whereas thermophilic ones do appear rigid [25,26,30]. However, this picture is not strictly correct or complete. As indicated by the universality of the calculated VDOS [111], the N-state of thermophilic enzymes is surely characterized by low frequency and long wavelength vibrational normal modes also at room temperature, whereas its functional high frequency and small wavelength vibrational normal modes are not populated at room temperature, but become sufficiently populated and excited only at higher temperature (note that RT ≈ 2.2 kJ mol^−1^ at 270 K, 2.5 kJ mol^−1^ at 300 K, and 3.1 kJ mol^−1^ at 373 K). In contrast, the N-state of psychrophilic enzymes has to be characterized by a sufficient population and excitation of functional high frequency and small wavelength vibrational normal modes also at very low temperatures. This fundamental dynamic property should be the consequence of subtle structural features, such as the absence of a strong coupling between the different parts of the folded structure for the presence of a higher fraction of residues with small side chains. The latter is emerged as a specific fingerprint of psychrophilic proteins in large-scale comparative investigations [27]. Clearly, relevant structural features can be more subtle and not so simple to unveil. For instance, cold adaptation driven by dynamic allostery (i.e., mutations in distant regions affect the functioning of the catalytic site) has been detected by means of both experimental and computational techniques in adenylate kinase [114], and malate dehydrogenase [115]. Now, we focus our attention on three specific enzymes: α-amylase (a monomer), citrate synthase (a dimer), and β-galactosidase (a tetramer).

### 4.1. α-Amylase

The family of cold-adapted enzymes more carefully investigated is surely that of chloride-dependent α-amylase. The latter enzyme is involved in the initial phase of starch digestion, specifically hydrolyzing starch into glucose. The research group of Feller and Gerday at the University of Liege has performed an in-depth characterization of the α-amylase from the psychrophilic microorganism *P. haloplanktis*, solving its 3D structure by means of X-ray diffraction [116,117], measuring its conformational stability by means of CD and DSC measurements, and characterizing its enzymatic properties [79,95,118,119]. A comparison between the k_cat_ values at 20 °C of the protein from *P. haloplanktis* and of the mesophilic protein from *Sus scrofa* is shown in Figure 7, highlighting the marked difference in enzymatic activity, regardless of the substrate, at low temperature.

A careful comparison with mesophilic and thermophilic homologues has also been carried out, together with the production and characterization of several mutant forms [120,121]. The psychrophilic enzyme showed T_d_ = 44 °C, both the mesophilic ones from *D. melanogaster* and *S. scrofa* showed T_d_ = 58 °C, the thermophilic one from *Thermobifida fusca* showed T_d_ = 77 °C [120]. Apart from the large difference in T_d_ values, DSC measurements indicated that the temperature-induced denaturation of the psychrophilic enzyme is a cooperative, reversible two-state transition, whereas it is characterized by the presence of intermediate states in the case of mesophilic and thermophilic α-amylases [120]. The psychrophilic enzyme has maximal activity at 29 °C, and k_cat_ = 179 s^−1^ at the physiological temperature of 5 °C; the mesophilic one from *D. melanogaster* has maximal activity at 54 °C, and k_cat_ = 280 s^−1^ at the physiological temperature of 20 °C; the mesophilic one from *S. scrofa* has maximal activity at 54 °C, and k_cat_ = 518 s^−1^ at the physiological temperature of 37 °C; the thermophilic one from *T. fusca* has maximal activity at 72 °C, and k_cat_ = 1457 s^−1^ at the physiological temperature of 55 °C (in all cases the enzymatic activity has been measured using 1% soluble starch as substrate). These numbers underscore that the “corresponding states” hypothesis in the case of chloride-dependent α-amylases, even though qualitatively correct, is not quantitatively right. Since k_cat_ increases exponentially with temperature (according to the Arrhenius equation), and the activation energy increases markedly on passing from the psychrophilic to the thermophilic enzyme [119], the latter possesses a k_cat_ larger than that of the cold-adapted enzyme at the physiological temperature of the respective microorganisms. In all probability, the thermophilic chloride-dependent α-amylase is more efficient than the psychrophilic homologue because, at high temperatures, the availability of more thermal energy stored in the vibrational normal modes allows a marked increase in the reaction rate. Feller and colleagues have also engineered several point mutations in the psychrophilic α-amylase based on the structure of the mesophilic homologue from *S. scrofa*. The inserted additional interactions (i.e., H-bonds, salt-bridges, and van der Waals contacts) caused both an increase in the thermal stability of the mutant proteins, and a significant decrease of the catalytic activity at low temperature [121]. These findings confirm the strict relationships existing between conformational stability, chain dynamics, and enzymatic activity [122].

### 4.2. Citrate Synthase

The enzyme citrate synthase (CS) catalyzes the entry of carbon, as acetyl-CoA, into the citric acid cycle, and occurs in almost all organisms. CS has a characteristic dimeric structure [123,124]; each monomer consists of a small and a large domain, with the central part of the dimer formed by the association of the two large domains. In each monomer, the gap between the two domains forms an active site that binds oxaloacetate first, followed by acetyl-CoA, leading to citrate formation. A substantial movement of the small domain with respect to the large domain is involved in the catalytic function, closing and opening the flaps, as the substrate binds to the active site slot. The research group of Danson and Taylor at the University of Bath solved the 3D structure of five CSs from psychrophilic, mesophilic, and thermophilic sources [123,125,126,127]. Specifically, they solved the structures of: (a) the enzyme from the psychrophilic *Arthrobacter DS2–3R*, whose optimum growth temperature is 31 °C, *Ar*CS; (b) the enzyme from pig, whose optimum growth temperature is 37 °C, *Pig*CS; (c) the enzyme from *Thermoplasma acidophilum*, whose optimum growth temperature is 55 °C, *Ta*CS; (d) the enzyme from *Sulfolobus solfataricus*, whose optimum growth temperature is 85 °C, *Ss*CS; (e) the enzyme from *Pyrococcus furiosus*, whose optimum growth temperature is 100 °C, *Pf*CS. The native structure of the five CSs proves to be very similar: the root mean square deviation between the Cα atoms for the complete dimeric structures solved via X-ray diffraction is at most 2.3 Å, even though the percentage of sequence identity is not large (i.e., 27% between *Ar*CS and *Pig*CS, 40% between *Ar*CS and *Pf*CS). The native structure of the five CSs possesses practically the same WASA, but there is a clear difference in the percentage of exposed hydrophobic WASA: the latter amounts to 29% in psychrophilic *Ar*CS, 20% in mesophilic *Pig*CS, and 18% in hyperthermophilic *Pf*CS. There are a lot of ionic interactions (i.e., both isolated ion pairs and charge-charge networks) in both psychrophilic *Ar*CS and thermophilic *Ss*CS and *Pf*CS. In the former, they occur mainly on the monomer surface, whereas, in the latter, they occur mainly at the interface between the two monomers. It appears that the stabilization of the dimer interface via ionic interactions is important to increase the thermal resistance in thermophilic CSs; in contrast, the significant presence of ionic interactions in the psychrophilic *Ar*CS should not be important for the stability of the native state, but it should play an important role for protein solubility in aqueous solution at low temperature [127]. The non-trivial relationship between enzymatic activity and thermal stability was confirmed by performing point mutations in the active site region of *Ar*CS, inspired by the structure of the hyperthermophilic *Pf*CS. On reducing the active site accessibility, the enzymatic activity at low temperature does not decrease and the thermal resistance increases [124].

### 4.3. β-Galactosidase

The enzyme β-galactosidase catalyzes the hydrolysis of lactose into its constituents monosaccharides, glucose, and galactose [128], and has attracted the interest of both researchers and the dairy industry due to nutritional, technological, and environmental problems associated with this important carbohydrate [129]. It is also used as a medical enzyme for producing Galalcto-Oligosaccharides, important for intestinal health [130]. The low-temperature catalytic activity of psychrophilic β-galactosidase is of relevant interest for potential industrial applications in diminishing lactose levels to avoid disorders in lactose-intolerant people [131]. Other applications are aimed at the preservation of heat-labile compounds [132,133]. A β-galactosidase was characterized by a strain of the Antarctic bacterium *P. haloplanktis* by the research group of Feller and Gerday at the University of Liege [134]. The psychrophilic microorganism shows the maximum production of β-galactosidase at around 4 °C. Such psychrophilic β-galactosidase proves to be very similar from the structural point of view with the mesophilic β-galactosidase from *E. coli*. The two enzymes have a tetrameric structure with a comparable subunit mass (each subunit consists of more than 1000 residues); they share 51% of sequence identity, the strict requirement for divalent metal ions (i.e., in line with the presence of two bound Mg^2+^ ions per monomer in the structure of the *E. coli* enzyme [135]), and a strong conservation of the amino acid residues constituting the active site. However, at 25 °C, the catalytic constants towards the natural substrate lactose are dramatically different: k_cat_ = 33 s^−1^ for the enzyme from *P. haloplanktis*, and 2 s^−1^ for the enzyme from *E. coli*. In addition, upon 50 min of incubation at 4 °C, the psychrophilic enzyme is able to hydrolyze 33% of milk lactose, whereas the mesophilic one hydrolyzes only 12% of milk lactose. The surprise is that the thermal stability of the two proteins is not so different: the denaturation temperature is 50 °C for the enzyme from *P. haloplanktis* and 57 °C for the enzyme from *E. coli*. This confirms that the temperature dependence of the conformational stability and that of the enzymatic activity are governed by different rules. It is worth noting that recently: (a) a hexameric β-galactosidase has been isolated from the psychrophilic *Marinomonas ef1*, and has shown remarkable stability against temperature, notwithstanding its highest activity at 5 °C [136]; (b) a dimeric cold-active β-galactosidase has been isolated from *Arthrobacter* sp. *32cB*, and is considered of relevant interest for food industry [137,138].

## 5. Conclusions

Psychrophilic enzymes represent a challenge to try to rationalize the relationship between their high catalytic activity at low temperature and their not-so-large conformational stability. The present analysis has offered some insight into the thermodynamics of the conformational stability of globular proteins, linking the marginal stability of the N-state to its large structural fluctuations. The latter have been discussed in the light of the harmonic approximation to describe the vibrational degrees of freedom of a molecule. In general, the enzymatic activity is associated with small wavelength vibrational normal modes localized in the active site region. Psychrophilic enzymes possess a sufficient excitation of such small wavelength vibrational normal modes to render operative the catalytic function at very low temperatures. This property should come from specific structural features that are difficult to decipher in view of the strong similarity existing between the native structure of homologous proteins from psychrophiles, mesophiles, and thermophiles.

## Figures and Tables

**Figure 1 microorganisms-10-01161-f001:**
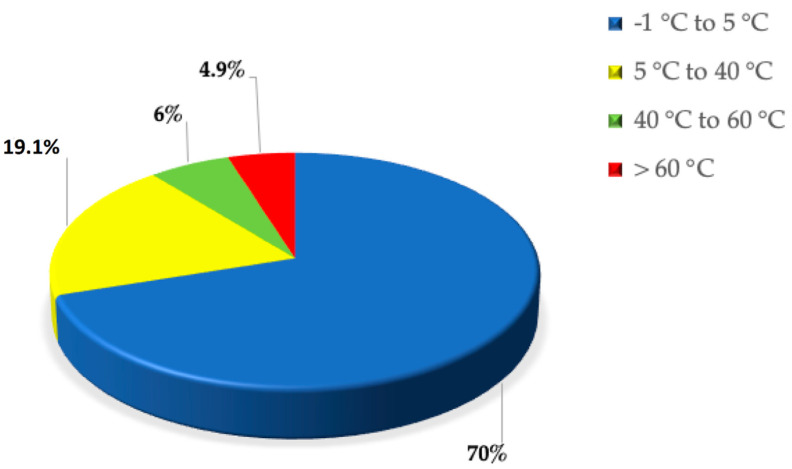
Pie–plot of Earth temperature distribution.

**Figure 2 microorganisms-10-01161-f002:**
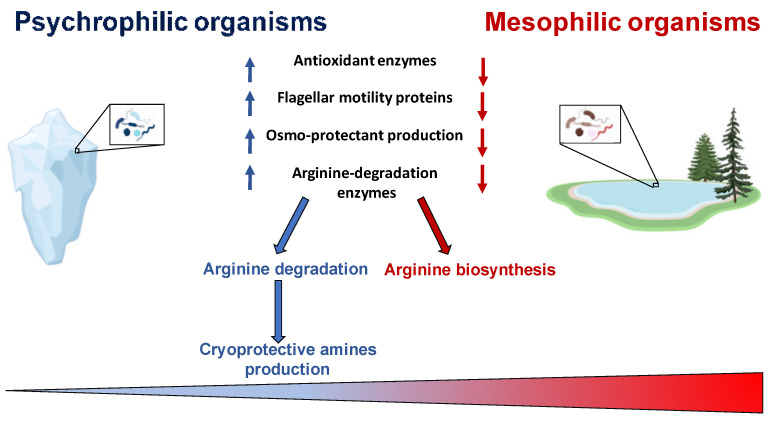
Metabolic adaptation. Image created with a template from BioRender.com.

**Figure 3 microorganisms-10-01161-f003:**
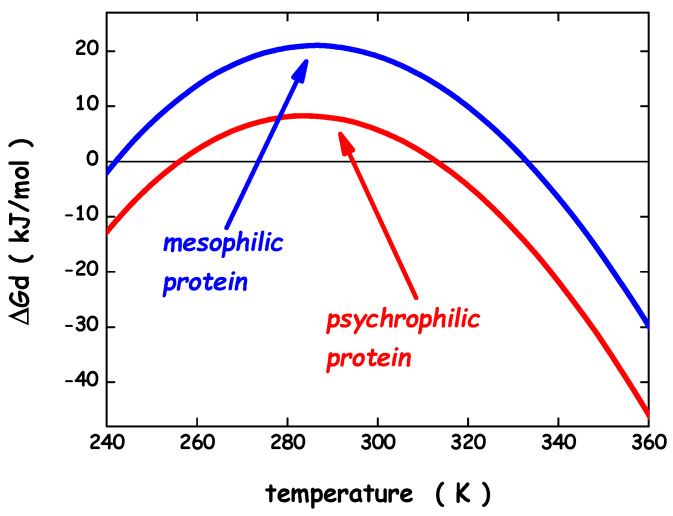
Thermodynamic stability curves for a model psychrophilic protein and a model mesophilic one. See text for further details.

**Figure 4 microorganisms-10-01161-f004:**
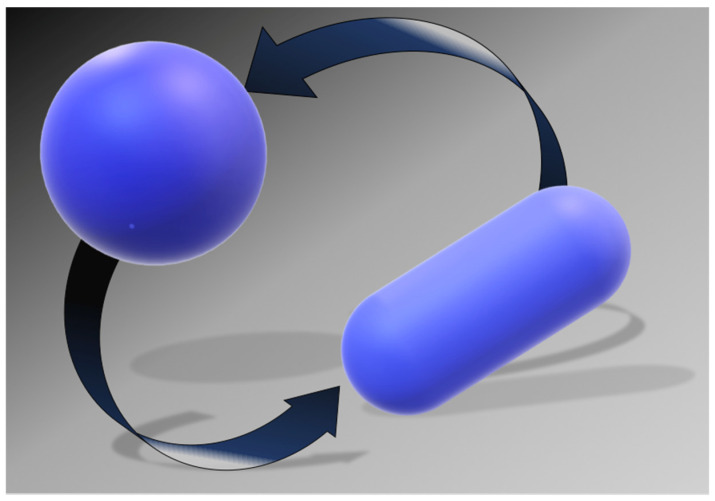
Simple geometric models for the two states of a globular protein: the sphere for the N-state and the prolate spherocylinder for the D-state.

**Figure 5 microorganisms-10-01161-f005:**
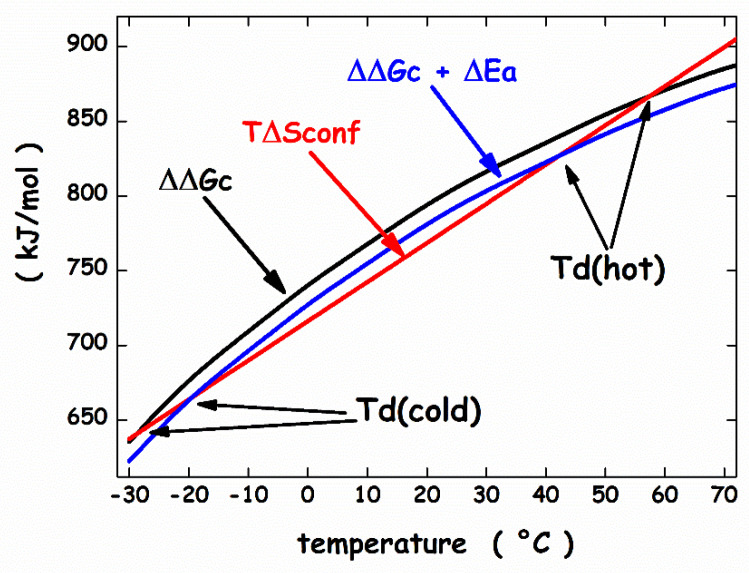
Temperature dependence of the functions, ΔG_c_, ΔΔG_c_ + ΔE_a_, and T⋅ΔS_conf_, calculated as described in the text.

**Figure 6 microorganisms-10-01161-f006:**
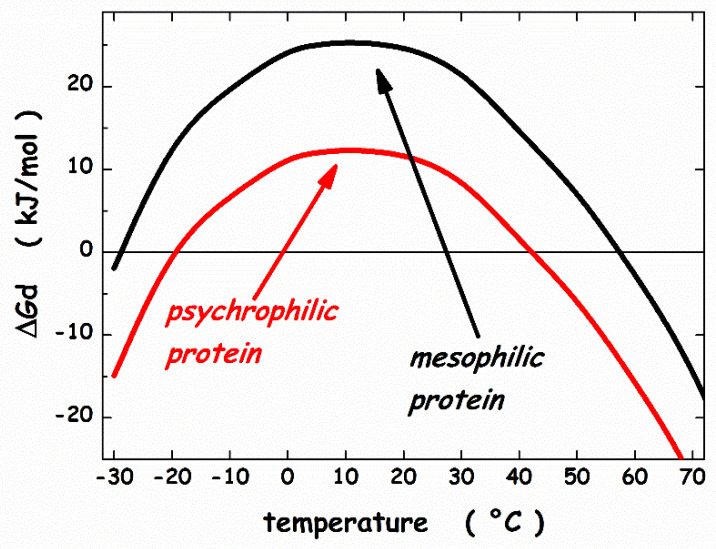
Thermodynamic stability curves calculated by means of the geometric theoretical approach. See text for further details.

**Figure 7 microorganisms-10-01161-f007:**
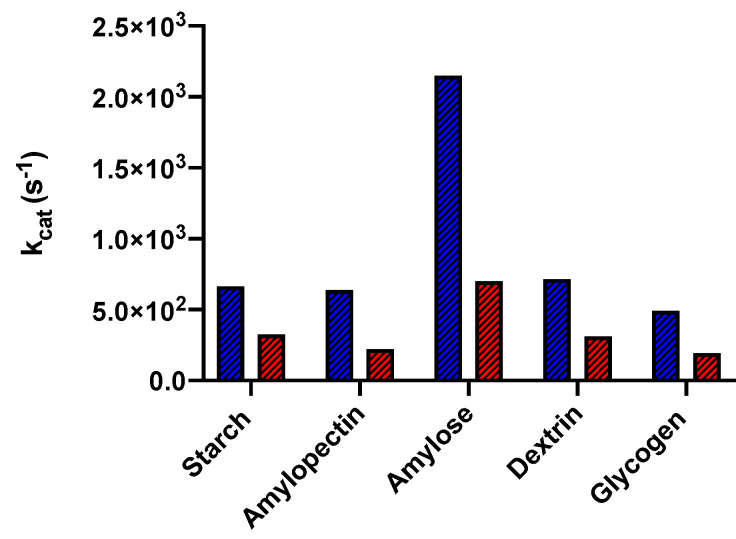
Graphical representation of k_cat_ values, at 20 °C, of α-amylase from the psychrophile *P. haloplanktis* (in blue) and from the mesophile *S. scrofa* (in red) on different substrates.

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
