# Peer review of "Some Clues about Enzymes from Psychrophilic Microorganisms"

_microorganisms, 2022, doi:10.3390/microorganisms10061161_

Round 1

Reviewer 1 Report

A review of Some clues about enzymes from psychrophilic microorganisms: A few points deserve attention for further publication. I suggest that it is accepted for publication after the following revisions:

 - The authors could clarify in the abstract of the manuscript the mechanism, advantages, problems, and solutions for the enzymes from psychrophilic microorganisms systems.

 - The authors should also highlight the advantages/disadvantages of these enzymes from psychrophilic microorganisms methods for industrial application and how this information will be addressed in the manuscript.

- Advantages for the enzymes from psychrophilic microorganisms systems: Which methods have advantages? Are they simple methods of contribution? When compared with other sustainable techniques? Authors need to leave this clear information throughout the text and the methods discussed in this manuscript. In addition, this information is needed for the enzymes from psychrophilic microorganisms systems contribution protocols to be applied on an industrial scale.

- Problems with the enzymes from psychrophilic microorganisms systems: Does this protocol have a significant problem? This discussion could be improved.

- Additionally, advances in the enzymes from psychrophilic microorganisms systems with engineered tailor-made have been performed with other strategies. May open new opportunities. This discussion could be improved.

- This review had broached interest in the progress and recent applications of the enzymes from psychrophilic microorganisms: The main contributions to the accomplishment of this work must be included in the conclusion.

 - Please, check all references according to the author's instructions.

 - The manuscript must be formatted according to the journal's standards.

Author Response

Reviewer 1 favours publication, but states that “few points deserve attention” and makes a small list that has a common topic: the potential advantages-disadvantages that psychrophilic enzymes offer for industrial and biotechnological applications. The topic is surely important and merits attention, but this is not the focus of our review. My research activity is mainly devoted to the rationalization of the conformational stability of globular proteins with its subtleties, trying to arrive at a reliable and consistent scenario (I have published more than 200 articles in peer-reviewed journals). I do not work on the industrial applications of enzymes from extremophiles. These sentences should clarify why the present review article is devoted to the globular protein aspects and features that I know in depth. The idea is to provide a different point of view on the intrinsic flexibility of psychrophilic enzymes and so to generate interest in the readers. This is why we have only partially addressed the points raised by Reviewer 1. In any case, we have added some sentences at the end of the Introduction with the relevant references.

Reviewer 2 Report

The review manuscript by Rapuano and Graziano focused on psychrophilic proteins. The review provide an insight on the thermodynamics of the conformational stability of these proteins and links the marginal stability to the large structural fluctuations observed.

The manuscript is nicely written and easy to read. However, I have a couple of major issues. Firstly, a check with Turnitin returns a match of 30% similarity. There are many passages taking verbatim from the authors previous publications which is not acceptable. The other issue is the lack of new or recent references. Is true that pioneer and relevant previous work deserve citation but this review relies heavily on them and such miss the opportunity to provide new findings/view on the subject. There is a large number of research and review articles on psychrophilic proteins published alone in the last 5 years which are not included in this manuscript. 

Other issues:

  • Abstract: alpha and beta symbols missing for amylase and galactosidase, respectively
  • L37-37. "There is an increasing interest in studying psychrophilic microorganisms to discover their secrets." What secrets?
  • L44-45. "Environmental adaptations allow these organisms to effectively avoid certain aspects of the cold environment" What certain aspects?
  • L75-76. Should read " following common rules hold for psychrophilic proteins" 
  • L84-85. "of the conformational stability of globular proteins in general, with a specific application of the so-called geometric model (developed by one of us) to psychrophilic proteins. Use references here [50,51?] instead one of us.
  • L100. "up-regulation of genes coding for cold shock proteins, CSP." Change to shock proteins (CSP)
  • L102. "are constitutively expressed and are classified as cold acclimatization proteins, CAP. Change to cold acclimatization proteins (CAP)
  • L146-147. "small globular proteins (i.e., single domain globular proteins; a structural domain should consist of less than 200 residues [41] is a reversible and cooperative process...The closing parenthesis is missing
  • L204-206. "the conformational stability of globular proteins. One of us has devised a geometric theoretical approach that works well in rationalizing the general occurrence of cold dena turation [50,51]. Why keep mentioning one of us rather than the authors or referencing references 50,51? 
  • L425. "Apart from the large.."
  • L460. "the thermophilic one from T. fusca"
  • The enzyme citrate synthase (CS)
  • Figure 2. Psychrophilic organisms (in blue) and  Mesophilic organisms (in red) are covered by other elements of the figure. Bottom of figure  "Cryoprotective amines production", production is cut-off at the bottom.

Author Response

Reviewer 2 favours publication of the manuscript, and raises two major points: (a) the percentage of similarity with my previous articles; (b) the need to make reference to more recent articles. A significant re-wording of several paragraphs has been done, as it can readily be verified. Some new references have been added, also to fulfil the suggestions of Reviewer 1. However, since the idea has been to provide a general view of the conformational stability of globular proteins with a basic connection to the vibrational normal modes of the folded state and to the role they play in enzymatic activity (trying to distinguish between psychrophilic enzymes and mesophilic-termophilic homologues), there is no strict need to make reference to recent articles, unless the latter provide new insight on the matter. In any case, relevant and recent references were already present in the original version of the manuscript: look for instance at references 86, 87 and 88 (in the actual list) published in Scientific Reports 2015, Nature 2018 and PNAS 2018. Moreover, all the minor points suggested by Reviewer 2 have been corrected.

Round 2

Reviewer 2 Report

Thank you for addressing some of the issues encountered in the original version. However, you have not addressed the Turnitin (plagiarism detection service) issue. Turnitin returned again a match of 28% similarity with many passages taking verbatim from the authors previous publications. This was commented in the initial review and has not been fully addressed by the authors. Please see attached Turnitin report of your revised manuscript.

Author Response

We have strongly modified the text of several paragraphs, as requested by the Reviewer 2, following the Turnitin results sent to us. We have checked the plagiarism of our modified text using the following program https://smallseotools.com/plagiarism-checker/  (we don’t have access to Turnitin). The check has found no plagiarism.

In addition, we have added more than 30 new references, all of them have been published in the last few years, and all of them are relevant to the matter of our manuscript. The new references are highlighted in yellow in the modified version of the manuscript. I have deleted no one of the references to my articles because I think they are relevant.